# Genomics in Pancreas–Kidney Transplantation: From Risk Stratification to Personalized Medicine

**DOI:** 10.3390/genes16080884

**Published:** 2025-07-26

**Authors:** Hande Aypek, Ozan Aygormez, Yasar Caliskan

**Affiliations:** 1Division of Nephrology, Bursa Uludag University School of Medicine, Bursa 16059, Türkiye; handeaypek@uludag.edu.tr; 2Division of Nephrology, Istanbul Faculty of Medicine, Istanbul University, Istanbul 34093, Türkiye; ozanaygormez@gmail.com; 3Division of Abdominal Transplant, SSM Health Saint Louis University Hospital, Saint Louis, MO 63104, USA

**Keywords:** simultaneous pancreas–kidney (SPK) transplantation, non-HLA genetics, pancreas transplantation, genomics

## Abstract

**Background:** Pancreas and pancreas–kidney transplantation are well-established therapeutic options for patients with type 1 diabetes mellitus (T1DM) and end-stage kidney disease (ESKD), offering the potential to restore endogenous insulin production and kidney function. It improves metabolic control, quality of life, and long-term survival. While surgical techniques and immunosuppressive strategies have advanced considerably, graft rejection and limited long-term graft survival remain significant clinical challenges. **Method:** To better understand these risks, the genetic and immunological factors that influence transplant outcomes are examined. Beyond traditional human leukocyte antigen (HLA) matching, non-HLA genetic variants such as gene deletions and single-nucleotide polymorphisms (SNPs) have emerged as contributors to alloimmune activation and graft failure. **Result:** Polymorphisms in cytokine genes, minor histocompatibility antigens, and immune-regulatory pathways have been implicated in transplant outcomes. However, the integration of such genomic data into clinical practice remains limited due to underexplored gene targets, variability in study results, and the lack of large, diverse, and well-characterized patient cohorts. Initiatives like the International Genetics & Translational Research in Transplantation Network (iGeneTRAiN) are addressing these limitations by aggregating genome-wide data from thousands of transplant donors and recipients across multiple centers. These large-scale collaborative efforts aim to identify clinically actionable genetic markers and support the development of personalized immunosuppressive strategies. **Conclusions:** Overall, genetic testing and genomics hold great promise in advancing precision medicine in pancreas and pancreas–kidney transplantation.

## 1. Introduction

Pancreas–kidney transplantation is increasingly recognized as a viable treatment option for patients with type 1 diabetes mellitus (T1DM) who develop end-stage kidney disease (ESKD). The principal modalities include simultaneous pancreas–kidney (SPK) and pancreas after kidney (PAK) transplants, each offering unique benefits related to preserving kidney function and managing diabetic complications. Compared to pancreas transplantation alone (PTA), the SPK transplant approach has shown superior outcomes, with studies showing prolonged graft survival and improved quality of life attributable to the synergistic protective effects on kidney function [1,2]. Furthermore, simultaneous transplantation may reduce immunological rejection risk compared to pancreas transplant alone or pancreas after kidney transplant, primarily because both organs are transplanted from the same deceased donor, resulting in identical HLA and non-HLA genetic mismatches for both grafts [1,2,3]. In parallel with clinical advances, recent progress in genomic research and high-throughput technologies has led to the identification of additional genetic factors implicated in long-term allograft function [4]. The discovery of genetic markers associated with graft loss offers the potential to refine risk stratification, optimize donor organ selection, and enhance individualized patient management [5,6,7,8,9,10]. For instance, the incorporation of genetic markers may facilitate the identification of marginal donor organs that possess favorable long-term functional potential, which might otherwise be declined based on conventional clinical or procurement assessments. Moreover, elucidating the molecular pathways linking specific genetic variants to allograft dysfunction may inform the development of novel therapeutic strategies or the repurposing of existing agents to mitigate graft injury and prolong allograft survival [11,12]. Importantly, the integration of genetic risk factors into clinical practice may enable the early identification of transplant recipients at elevated risk for graft failure, thereby allowing for intensified surveillance and preemptive interventions prior to the onset of irreversible graft injury [5,8,9].

In this narrative review, we will examine the current evidence regarding genetic determinants of long-term pancreas transplant outcomes, highlight the methodological challenges inherent in the identification and validation of these variants, and discuss emerging directions for the application of genetic predictors in the field of pancreas transplantation.

The scope of this study aims to provide readers with a contextual understanding of existing evidence and highlight areas where further research is needed.

## 2. Non-HLA Genetic Factors in Solid Organ Transplantation

The genetics of pancreas–kidney transplantation primarily concerns the influence of both donor and recipient genetic factors on graft outcomes, particularly long-term graft function and the risk of acute rejection. Genetic compatibility between the donor and recipient plays a critical role in the success of both pancreas and kidney transplants. Factors such as HLA matching contribute significantly to minimizing acute rejection episodes, particularly when both organs are derived from the same donor, which has been shown to enhance outcomes further [13,14]. Although HLA matching remains a cornerstone of immunogenetic compatibility assessment, recent data suggest that other genetic factors, including non-HLA genetic variants, may also influence graft survival and rejection risk.

Non-HLA genetic factors, including minor histocompatibility antigens, gene polymorphisms outside the HLA region, and immune responses directed against non-HLAs, are increasingly recognized as important contributors to kidney transplant outcomes. These factors can independently or synergistically contribute to the risk of graft rejection and loss, complementing the impact of classical HLA mismatches [5,15,16,17,18,19,20]. Studies have shown that specific non-HLA gene variants can provoke alloimmune responses similar to those triggered by HLA mismatches, leading to acute rejection and poor graft survival [15,16,17,18,19,20,21]. Polymorphisms in non-classical HLA molecules such as major histocompatibility complex class I-related chain A (MICA) and HLA-G, as well as the presence of antibodies against these antigens, have been linked to adverse transplant outcomes [18,22,23]. Additionally, immune responses against minor histocompatibility antigens, such as H-Y antigens encoded on the Y chromosome (e.g., MEA1), may contribute to worse graft survival, particularly in female recipients of male donor kidneys [20,21,22,23,24]. Genome-wide association studies (GWASs) and large sequencing consortia like the International Genetics & Translational Research in Transplantation Network (iGeneTRAiN) have further elucidated the role of non-HLA genetics in transplantation [25,26,27,28,29]. Recent genome-wide analyses have demonstrated that mismatches in transmembrane and secreted protein-coding genes between donors and recipients are independently associated with graft loss, even after adjusting for HLA eplet mismatches [29]. Moreover, non-HLA and HLA responses may act synergistically, complicating the immunological risk landscape [30].

The integration of genetic considerations into the transplant process, primarily by ensuring optimal donor–recipient matching, greatly enhances the outcomes of patients undergoing SPK transplantation. This combination significantly improves graft survival rates and provides protective benefits against the long-term complications associated with diabetes, marking it a critical area of focus for both surgical and transplant medicine. Additionally, genetic markers are being investigated for their potential to predict long-term graft function. These markers may help in donor organ selection, understanding the mechanisms of graft loss, and identifying patients at risk for graft failure before clinical deterioration occurs (Figure 1). However, while several genetic variants have been associated with outcomes, their clinical utility remains investigational, and no specific genetic screening is currently standard in clinical practice for pancreas–kidney transplantation [31]. Despite advances in the field, the most significant genetic risk factors currently recognized in clinical pancreas transplantation practice are related to HLA mismatching and recipient immune response gene polymorphisms [31,32].

## 3. Recipient and Donor Non-HLA Genetics in Pancreas–Kidney Transplantation

### 3.1. Recipient Genetics in Pancreas–Kidney Transplantation

Recipient genetics influence the outcomes of pancreas and pancreas–kidney transplantation through various mechanisms that impact graft survival, the risk of complications, and overall patient health. There are genetic factors such as polymorphisms in cytokine genes, HLA alleles, and other immune-related genes [32,33,34]. Besides HLA mismatches, non-HLA genetic factors are thought to contribute to the immunological response and transplant outcomes. Understanding these genetic factors can help optimize pre-transplant assessments, immunosuppressive strategies, and the long-term management of transplant recipients [31]. This section will cover the genetics of pancreas diseases and provide an overview of the non-HLA genetics of pancreas and pancreas–kidney transplantation recipients.

#### 3.1.1. Monogenic Diseases of Pancreas and Transplantation

Although monogenic diseases of the pancreas are rare, several types have been identified, with maturity-onset diabetes of the young (MODY) and neonatal diabetes being the most studied. MODY has several subtypes defined by the causative gene mutation [35]. Specifically, mutations in the *HNF1A*, *HNF4A*, and *GCK* genes are known to cause MODY [36]. On the other hand, neonatal diabetes often results from mutations in genes such as *KCNJ11* and *ABCC8*, which encode components of the ATP-sensitive potassium channel in pancreatic beta cells [37].

Studies on MODY subtypes indicate the effects of genetic mutations on pancreatic function in transplant recipients. The clinical features and post-transplant outcomes of patients with MODY3, caused by *HNF1A* mutations, and Renal Cysts and Diabetes syndrome (RCAD), often linked to *HNF1B* mutations, who undergo kidney and pancreas transplantation, suggest that transplantation is a viable treatment option for these patients. However, close post-transplant monitoring is essential due to the distinct genetic and metabolic profiles associated with these monogenic forms of diabetes [38]. Although often underdiagnosed, MODY may be present in a subset of pancreas transplant candidates, particularly those with a history of atypical diabetes, preserved C-peptide, and a family history of diabetes or renal disease [38,39].

Pancreas transplantation has been shown to be effective in selected MODY3 and RCAD patients, with good graft and patient survival rates comparable to those seen in other forms of diabetes mellitus. In a single-center experience, MODY3 and RCAD patients who underwent pancreas and/or kidney transplantation had functioning grafts at follow-up, supporting the benefit of transplantation in appropriately selected cases. However, MODY patients often have persistent C-peptide secretion and lack autoantibodies, which may influence both candidacy and expected outcomes. For example, some MODY patients may retain sufficient endogenous insulin secretion to avoid transplantation altogether, or may respond to oral hypoglycemic agents, particularly sulfonylureas, in *HNF1A*-MODY. Careful patient selection is therefore critical to maximize benefits and avoid unnecessary transplantation in those unlikely to benefit, such as those with mild hyperglycemia or preserved beta-cell function [39,40,41].

Misclassification of the type of diabetes mellitus can lead to inappropriate therapies. For example, patients with MODY misdiagnosed as T1DM may undergo unnecessary pancreas transplantation or be exposed to lifelong immunosuppression when they could be managed effectively with oral agents [39,42,43]. Genetic diagnosis of MODY enables precision medicine: it guides the selection of candidates who are most likely to benefit from transplantation, informs the choice of immunosuppressive regimen, and prevents the use of ineffective or potentially harmful interventions [38,39,42,43].

Mutations in the *INS* gene, including both dominant negative coding mutations and novel intronic variants, disrupt pancreatic beta cell identity and insulin gene expression, leading to permanent neonatal diabetes. These types of mutations can often escape conventional diagnoses, but have significant implications for pancreatic transplantation, making genetic testing essential for optimal outcomes [44,45].

A study on pediatric organ transplant recipients in Italy found that over 90% had rare diseases, with more than 60% of these being monogenic disorders. It showed the significance of genetic factors in pediatric transplantation and underlined the need for management strategies [46].

Two rare cases related to recipient genetic factors have been identified in pancreas transplantation. This included a child with Wolcott–Rallison syndrome who underwent combined liver, pancreas, and kidney transplantation due to recurrent liver and kidney failure. Wolcott–Rallison syndrome is a rare genetic disorder causing early-onset diabetes, liver failure, and skeletal abnormalities. The multi-organ transplant approach showed potential in managing severe disease manifestations, improving the patient’s survival, and quality of life [47]. The second case involves the challenges of managing immunosuppression and infection risk in a SPK transplant recipient with common variable immunodeficiency due to a *TNFRSF13B* gene mutation [48].

Despite these advances, significant gaps remain in the literature regarding the clinical application of genetic testing, long-term outcomes, and optimal immunosuppressive protocols for transplant recipients with MODY. Understanding the genetic condition of monogenic diabetes is crucial for optimizing transplant outcomes. Further research is needed to refine patient selection and management strategies in this unique population.

#### 3.1.2. Complex Genetic Diseases of Pancreas and Transplantation

Complex genetic pancreas diseases arise from the combined effect of multiple genetic factors and often interact with environmental determinants. T1DM and Type 2 diabetes mellitus (T2DM) are examples of complex genetic pancreas diseases [49,50] and will be outlined in this review. T1DM has a complex genetic basis involving multiple immune-regulating genes, particularly within the HLA region, which contribute to autoimmune pancreatic beta-cell destruction, while T2DM arises from a combination of genetic and environmental factors affecting insulin secretion and metabolism [51,52]. Although pancreas transplantation has traditionally targeted T1DM patients, growing insights into the genetic and metabolic characteristics of T2DM are expanding its applicability, potentially improving graft outcomes and overall success [53,54,55]. The understanding and management of these diseases can influence transplantation outcomes, including graft viability, functional success, and complication rates.

In T1DM patients, specific HLA alleles correlate with an increased risk of developing diabetes, and autoimmune recurrence after pancreas transplantation poses a significant challenge, as type 1 diabetes can reappear in the graft even without detectable glutamic acid decarboxylase (GAD) and insulinoma-associated protein 2 (IA-2) autoantibodies [56,57,58]. Early detection is a critical step for managing recurrence and improving long-term graft survival.

### 3.2. Donor Genetics in Pancreas–Kidney Transplantation

While the recipient’s immune system has traditionally been the main focus in transplantation research, recent studies suggest that the genetic profile of the donor can also play a significant role in long-term graft outcomes, especially in SPK transplantation. Donor-specific genetic variants may influence how the recipient’s immune system responds, how well the organ tolerates stress, and even whether autoimmune processes return after the transplant.

One of the key genes of interest is Caveolin-1 (*CAV1*), which is involved in cellular signaling and tissue remodeling. Certain *CAV1* gene variants, such as rs3801995 and rs9920, have been linked to higher rates of fibrosis in kidney allografts, and similar mechanisms may also affect pancreas transplants [59]. These findings suggest that the donor’s *CAV1* genotype might be a useful indicator of long-term organ health and could help guide the use of marginal organs.

In addition to structural genes, metabolic genes in the donor may also impact transplant outcomes. For example, variants in the Apolipoprotein E (*APOE*) gene, especially the ε4 allele, are associated with higher cholesterol levels and cardiovascular risk after transplantation [60]. However, since SPK transplantation often improves metabolic control, the long-term impact of these donor variants may be reduced.

Immune regulatory genes like the vitamin D receptor (*VDR*) also show promise. Some early studies suggest that specific *VDR* genotypes in the donor, such as *FokI* FF, might be associated with a higher risk of immune rejection, although further studies are needed [61]. Similarly, donor variants in the Heme Oxygenase-1 (*HO-1*) gene, a key enzyme that protects against oxidative stress, may influence how well the organ tolerates ischemia and reperfusion injury [62]. Donors with “high-expression” *HO-1* genotypes appear to offer more resilient grafts, particularly under stress conditions. Experimental models further suggest that the pharmacologic upregulation of HO-1 prior to transplantation can enhance graft survival and function by reducing inflammatory responses and protecting against ischemia/reperfusion-induced tissue injury [63].

Beyond genetic polymorphisms, other donor-related factors may shape transplant success. For instance, sex mismatch between the donor and recipient has been studied as a possible risk factor. Some reports suggest that female recipients of male donor organs may face a higher risk of graft loss, potentially due to immune responses against male-specific proteins (H-Y antigens) [64]. However, other studies have not confirmed this association [65], making it difficult to draw definitive conclusions.

There is also evidence that immune responses can be shaped by how the donor expresses certain non-HLAs. For example, the expression of MICA in the donor pancreas has been associated with the development of anti-MICA antibodies in the recipient, which can lead to rejection [66]. In patients with T1DM, SPK transplantation may even trigger a recurrence of autoimmunity. Some recipients develop or re-develop antibodies against pancreas-specific antigens like GAD65, IA-2, or ZnT8, particularly when diabetes comes back after transplantation [58,67].

Together, these findings highlight that the donor is not a passive element in transplantation. Understanding donor genetics, ranging from fibrosis-related genes to immune markers, can offer valuable insights into graft prognosis. As molecular tools become more widely used, incorporating donor genetic screening into clinical decision-making may help tailor post-transplant care and improve outcomes in SPK recipients.

## 4. Genetic Factors and Biomarkers for Pancreas Transplantation

Genetic factors and biomarkers are important and necessary for monitoring and managing pancreas transplantation, offering insights into graft function, rejection, and recovery. Advances in sequencing technology have enhanced the characterization of promising genetic factors as biomarkers for the early detection and assessment of graft status.

Regarding non-HLA genetic factors in pancreas–kidney transplantation, many studies have been identified (Table 1). Polymorphisms in cytokine genes such as *TNF-α* and *IL-10* in recipients are associated with an increased risk of acute rejection episodes after kidney [68,69,70] and SPK transplantation. Specifically, a high TNF-α production phenotype is correlated with recurrent acute rejection, and certain *IL-10* genotypes may also contribute to higher rejection rates, especially in the context of low TNF-α production phenotypes [32]. Genetic factors such as *APOE* and *VDR* polymorphisms may play a role in pancreas transplant outcomes. Individuals carrying the *APOE4* allele exhibit unfavorable lipid profiles pre-transplant, suggesting an increased cardiovascular risk [60]. Those with the FF genotype of the *VDR FokI* polymorphism appear more susceptible to acute rejection [61]. In addition, Factor V Leiden (*F5*) SNP (rs6025) showed a strong and validated association with acute rejection in SPK patients [71]. These findings emphasize the impact of genetic polymorphisms on pancreas transplant rejection and patient outcomes.

Immune profiling of peripheral blood mononuclear cells reflects acute pancreas rejection episodes by indicating elevated levels of circulating CD4+, CD8+ T cells, and CD19+ B cells in SPK transplant recipients [33]. SPK transplant recipients, including those experiencing biopsy-proven acute rejection, developed de novo antibodies against kidney-specific self-antigens (collagen-IV and fibronectin) while no significant response was observed against pancreas-specific antigens. Elevated kidney-specific autoantibodies correlated with heightened cytokine responses, suggesting that alloimmune-triggered autoimmunity may contribute to rejection in these patients [72]. The presence of pancreatic autoantibodies, such as anti-GAD and anti-insulin antibodies, before transplantation does not predict graft loss. However, the development or persistence of these autoantibodies after transplant is associated with poorer glycemic control and reduced β-cell function [67]. Together, these findings indicate the complex interplay between alloimmune responses and autoimmunity in SPK transplant rejection.

A pilot study used gene expression profiling to categorize pancreas transplant biopsies into groups reflecting either functioning grafts or rejection. The elevated expression of specific genes, such as *CD20*, was associated with acute and chronic rejection and correlated with poorer clinical outcomes [73]. Moreover, a 34-gene expression panel effectively identifies antibody-mediated rejection in pancreas transplants and predicts allograft failure, supporting its use as a valuable molecular diagnostic tool [74]. The Tissue Common Response Module (tCRM) score, derived from gene expression analysis, has been validated in pancreas transplants to assess rejection severity and resistance to treatment, aiding in the management of acute cellular rejection [75]. These studies demonstrate that gene expression profiling provides valuable molecular tools for diagnosing pancreas transplant rejection, predicting allograft failure, and guiding treatment strategies.

Low pre-transplant mannose-binding lectin (MBL) levels, potentially influenced by specific gene polymorphisms, are associated with improved patient and graft survival following SPK transplants [76]. Moreover, the *MBL2* genotype influences the normalization of MBL levels and their association with vascular endothelial growth factor (VEGF) expression in SPK transplantation [77], underlining MBL as a potential biomarker that specific gene polymorphisms and MBL levels could be correlated with in pre- and post-transplant risk assessments.

Granzyme B (*GNMZ*), perforin (*PRF1*), and *HLA-DR* gene expressions in peripheral blood show promise as early, non-invasive biomarkers for detecting and monitoring acute pancreas transplant rejection, aiding larger cohort studies [78,79,80] (Table 1). Elevated levels of donor-derived cell-free DNA (dd-cfDNA) in plasma have shown high sensitivity and specificity for diagnosing pancreas graft rejection, particularly beyond 45 days post-transplantation [81]. These biomarkers outperform traditional markers like lipase and amylase and may reduce the need for invasive biopsies.

Collectively, these biomarkers and approaches in the studies can enhance the ability to detect early rejection, monitor graft function, and ultimately improve patient outcomes in pancreas transplantation and guide the personalized immunosuppressive therapy of the recipients (Table 2).genes-16-00884-t001_Table 1Table 1Summary of studies on non-HLA genetic factors in pancreas transplantation.StudyStudy TypeTransplant Type and CohortPopulationGenes TargetsMethodsOutcomesMain FindingsStrengths and Limitations**Pelletier et al.** **[32]**Candidate gene associationSPK (n = 19), Kidney (n = 82)Patients at a single U.S. transplant centerTNF-α, IL-10, IFN-γ, TGF-β polymorphismsGenotyping of cytokine gene polymorphisms by polymerase chain reaction (PCR)Graft rejection (acute rejection incidence, recurrence, severity, and graft function)High TNF-α and IL-10 phenotypes linked to higher AR riskStrengths: functional genotyping; clearly defined outcomes. Limitations: small, single-center cohort study; lack of validation**Balakrishnan et al.** **[60]**Candidate gene association; prospective pre/post-transplant lipid assessmentSPK (n = 84), PTA (n = 9)Adult T1DM patients awaiting pancreas transplantation at a single U.S. transplant centerAPOE alleles: ε2, ε3, ε4Genotyping of ApoE alleles by PCR and restriction fragment length polymorphism (PCR-RFLP)Lipid profiles (Triglycerides, HDL-C, and cholesterol-to-HDL ratio before and after transplantation)E4 allele linked to higher TG, lower HDL, and a higher C/H ratio pre-transplant; no E2 effect; differences were resolved post-transplantStrengths: within-subject design; functional genotyping; clear lipid endpoints. Limitations: small size, single-center study; lack of multivariable adjustment; historic data**Hankey et al.** **[66]**Observational, descriptive study analyzing tissue expression of MIC antigens in human allograft biopsiesSPK (n = 10), Kidney (n = 19)Patients who received kidney or pancreatic transplants and underwent biopsy for clinical indicationsMICA, MICBIndirect immunohistochemistry (IHC) using a monoclonal antibody directed against MICA and MICBGraft rejection (MIC expression presence and localization correlated with histopathological diagnoses)MIC was absent/minimal in non-rejecting renal biopsies, but MIC was strongly expressed in acute and chronic rejection renal biopsies. MIC found in pancreatic biopsies with or without rejectionStrengths: specific monoclonal antibodies; inclusion of renal and pancreatic samples; use of controls; histological correlation. Limitations: descriptive design; limited sample size; no longitudinal or clinical outcome data; single-center study**Cashion et al.** **[78]**Observational gene expression studySPK (n = 3), PTA (n = 9), PAK (n = 3)Adult pancreas transplant recipients and controls with/without T1DMGNMZ, PRF1Quantitative real-time PCR (qRT-PCR) of peripheral bloodGraft rejection (gene expression levels correlated with biopsy-proven rejection)Higher granzyme B, perforin, and HLA-DRA levels in the rejection group (not statistically significant). Granzyme B was significantly higher vs. diabetic controls. Granzyme B may be useful for non-invasive rejection monitoringStrengths: precise gene expression measurement via qRT-PCR; inclusion of diabetic and non-diabetic controls. Limitations: small sample size; lack of statistical significance between rejection and non-rejection groups**Becker et al.** **[63]**Experimental animal studyPTA (n = 8–10 per group) *Inbred Lewis rats used as donors and recipientsHO-1Induction of HO-1 using cobalt protoporphyrin (CoPP) in donor rats prior to pancreas procurement; grafts stored in HTK solution and transplanted into recipientsGraft survival (endocrine and exocrine function, serum lipase activity, histopathological examination, HO-1 gene expression levels, and cytokine profiles)CoPP pretreatment resulted in 100% graft survival after prolonged cold ischemia, compared to 37.5% in controls. Enhanced HO-1 gene expression (130-fold increase) in the donor pancreas. Improved endocrine function and reduced serum lipase activity. Preservation of graft architectureStrengths: controlled experimental design with defined pretreatment and post-transplant assessments; comprehensive evaluation. Limitations: study conducted in a rat model; results may not directly translate to human pancreas transplantation**Luan et al.** **[73]**Pilot feasibility study evaluating mRNA-based stratification of pancreas transplant biopsiesSPK (n = 14), PAK (n = 12)Pancreas transplant recipientsCD20qRT-PCR of biopsy tissueGraft loss (gene expression profiles, cluster analysis, and correlation with clinical outcomes)Unsupervised 2D hierarchical clustering segregated biopsies into two main clusters: Cluster A: 85.7% graft survival; Cluster B: 31.6% graft survival. Detection of CD20/MS4A1 mRNA and protein in some biopsies in Cluster BStrengths: Utilization of archived pancreas biopsy specimens. Identification of potential molecular markers associated with graft survival. Limitations: small sample size and variability in immunosuppressive protocols over time**Cashion et al.** **[80]**Longitudinal observational studySPK (n = 4), PAK (n = 5), PTA (n = 4)Adult recipients of pancreas allograftsGNMZ, PRF1, HLA-DRAqRT-PCR analysis of peripheral blood mononuclear cellsGraft rejection (levels of granzyme B, perforin, and HLA-DRα mRNA; correlation with biopsy-proven acute rejection episodes; response to immunosuppressive therapy)A significant increase in biomarker levels was observed up to 5 weeks before clinical diagnosis of acute rejection. Biomarker levels decreased following intensified immunosuppressive therapy in all patients with biopsy-proven rejectionStrengths: Longitudinal design with multiple follow-up points. Direct correlation of biomarker levels with histologically confirmed rejection episodes. Limitations: Small sample. Variability in baseline biomarker levels among individuals**Oetting et al.** **[71]**Multi-center, observational cohort studySPK (n = 62), Kidney (n = 907)Clinically well-defined kidney transplant recipients from multiple centersF5Genotyping of 23 previously reported SNPs associated with acute rejection; statistical analyses including race-adjusted and multivariable modelsAcute rejection (association between SNPs and biopsy-proven acute rejection episodes)Only one SNP, rs6025 (Leiden mutation) in the coagulation Factor V gene, showed a significant association with AR (*p* = 0.011 in race-adjusted analysis; *p* = 0.0003 in multivariable analysis).Strengths: large, multi-center cohort and rigorous statistical analyses, including race-adjusted and multivariable models. Limitations: Study focused on a limited number of SNPs; other potentially relevant genetic variants were not assessed**Rahsaz et al.** **[61]**Pilot, observational studySPK (n = 7), PAK (n = 3), PTA (n = 11)Adult pancreas transplant recipients and healthy individuals from IranVDR (Fokl polymorphism)Genotyping of the vitamin D receptor (VDR) gene (FokI polymorphism)Acute rejection (association between VDR FokI genotype and incidence of acute rejection episodes)All patients with acute rejection had the FF genotype; no homozygous ff genotype was identified. The frequency of the FF genotype was higher in the rejection group compared to the non-rejection group (71% vs. 60%)Strengths: Focus on a specific genetic polymorphism. Comparison with a healthy control group to assess genotype distribution. Limitations: Small sample size limits statistical power. Lack of long-term follow-up to assess graft survival**Poitou et al.** **[38]**Single-center observational studySPK (n = 4), Kidney (n = 2)Patients with MODY3 and RCADHNF1A, HNF1BClinical evaluation, genetic testing, and follow-up post kidney and/or pancreas transplantDiabetes after transplant (transplant success, graft survival, and post-transplant metabolic control)All patients with MODY3 developed diabetic nephropathy, while only about half of the RCAD patients had diabetic kidney damage. Transplantation was safe and effective in MODY3 and RCAD patientsStrengths: Distinct clinical profiles for MODY3 vs. RCAD; both groups showed good transplant outcomes, with genetic diagnosis important for management. Limitations: single-center; small sample size**Martins et al.** **[67]**Observational cohort studySPK (n = 105)Patients undergoing SPK for T1DMGAD2Measurement of pancreatic autoantibodies (e.g., GAD, IA-2) pre- and post-transplant; clinical follow-upGlycemic control (correlation between autoantibody presence and graft function, rejection episodes, and graft survival)The presence or persistence of pancreatic autoantibodies post-transplant may not predict graft loss consistently; clinical relevance debatedStrengths: Longitudinal monitoring of autoantibodies. Clinical correlation with graft outcomes. Limitations: variability in autoantibody assays and timing**Hamilton et al.** **[59]**Genetic association study with clinical follow-upSPK (n = 315), PAK (n = 68), PTA (n = 38), SPT (n = 14)Pancreas transplant recipients, primarily with T1DMCAV1 (rs3801995, rs992)Genotyping for CAV1 polymorphisms; analysis of graft function and survival over timeGraft survival (association between CAV1 genetic variants and long-term pancreas graft function and survival)Specific CAV1 variants correlate with improved or reduced long-term pancreas transplant function, suggesting their genetic influence on graft outcomesStrengths: integration of genetic data with clinical transplant outcomes; focus on long-term function. Limitations: limited sample size**Gunasekaran et al.** **[72]**Observational cohort study with immunological analysisSPK (n = 39)Patients undergoing SPK transplantationCOL4A1, FN1, GAD, ICA, PAP-1Measurement of antibodies and T-cell responses against tissue-restricted self-antigens; clinical correlation with episodes of acute rejectionGraft rejection (association between immune response to self-antigens and acute rejection episodes; graft function)Development of immune responses to tissue-restricted self-antigens is associated with acute rejection in SPK recipients, suggesting its role in graft injuryStrengths: detailed immunological profiling, direct clinical correlation, and focus on novel antigen targets. Limitations: small sample size; single-center study**Roufosse et al.** **[74]**Molecular and histopathological observational studySPK (n = 29), PAK (n = 8), PTA (n = 4)Pancreas transplant recipients undergoing biopsy for suspected rejectionAMR 34-gene scoreMolecular profiling (e.g., gene expression analysis) of pancreas allograft biopsies; histological examination; correlation with clinical rejection dataGraft loss (challenges related to immunodeficiency impacting transplant success, infection risk, and graft function)Molecular markers can reliably detect AMR in pancreas allografts, improving diagnostic accuracy beyond traditional histology aloneStrengths: use of advanced molecular techniques; integration with clinical and histological data. Limitations: small biopsy sample size; single-center study**Coimbra et al.** **[48]**Case reportSPK (n = 1)Patients with common variable immunodeficiency (CVID) undergoing SPK transplantTNFRSF13B, BACH2 (VOUS)Genetic analysis; clinical monitoring of transplant outcomes; management of immunodeficiency-related complicationsGraft rejection (challenges related to immunodeficiency impacting transplant success, infection risk, and graft function)TNFRSF13B mutation-related CVID can complicate post-transplant management, increasing the infection risk and influencing graft outcomesStrengths: detailed genetic and clinical correlation in transplant context; highlights rare but important complications. Limitations: Single case report**Brown et al.** **[75]**Molecular observational study with gene expression profilingSPK (n = 14), PAK (n = 12)Pancreas transplant recipients undergoing biopsy for rejection assessment37 genes in Grade 3 ACR, 56 in Grade 2 ACRMolecular scoring based on gene expression analysis from allograft biopsy samples; correlation with histology and clinical outcomesRejection severity (quantitative molecular scores correlated with the severity of rejection and response to immunosuppressive treatment)Molecular scoring provides a sensitive, quantitative measure of rejection severity and can predict resistance to treatment, improving management decisionsStrengths: Objective molecular assessment, integration with clinical and histological data, and potential for personalized treatment. Limitations: small sample size; requires specialized molecular techniquesTransplant organ: PTA, pancreas transplant alone; SPK, simultaneous pancreas–kidney transplant; PAK, pancreas after kidney transplant; SPT, second pancreas transplant; AMR, antibody-mediated rejection; ACR, acute cellular rejection; VOUS, variant of uncertain significance. * Lewis rats were utilized as the experimental model in this study.


## 5. Non-HLA Genomic Mismatch and Challenges in Pancreas–Kidney Transplantation

### 5.1. Non-HLA Genomic Mismatch

Non-HLA genomic mismatch involves variations in genes encoding minor histocompatibility antigens, immune regulatory proteins, cytokines, and other molecules involved in inflammation, tissue repair, and immune recognition [20,29]. Non-HLA genomic mismatches between the donor and recipient have been identified as significant contributors to graft rejection and survival outcomes across various organ transplants. Advancements in genomic technologies have facilitated the identification of relevant non-HLA loci, suggesting that incorporating comprehensive genetic profiling beyond HLA matching may enhance transplant success and inform personalized immunosuppressive strategies in transplantation [82,83].

Even though the impact of non-HLA mismatches in pancreas transplantation remains underexplored, a couple of specific non-HLA genetic factors have been investigated for their roles in transplant outcomes. Variations in *MICA*/*MICB* genes and inflammatory cytokine genes such as TNF-α, IL-6, IL-10, and TGF-β have been associated with immune activity and transplant success [31,32,66]. In addition, specific *APOE* allele and *VDR* and *Cav1* polymorphisms are examples of non-HLA genomic mismatch in pancreas transplantation [59,60,61]. Genetic mismatches in non-HLA genes may influence the risk of both acute and chronic rejection, graft fibrosis, and overall long-term graft survival in pancreas transplantation. Ultimately, integrating non-HLA genetic data into clinical practice may enhance outcomes in pancreas transplantation and reduce the risk of graft loss [38]. Further research is needed to determine the effect of non-HLA mismatches on pancreas transplant rejection and survival.

### 5.2. Challenges in Pancreas–Kidney Transplantation

Although non-HLA genetics has attracted growing interest in the transplant field, several barriers remain before these findings can be widely applied in clinical practice. While research has uncovered a number of non-HLA genetic variants that seem to influence graft outcomes, we still face challenges related to consistency, interpretation, and implementation.

While a growing number of studies have identified genetic variants associated with graft outcomes in kidney and pancreas transplantation, it is important to emphasize that the current body of evidence remains largely exploratory. Most of the available data stem from small, often single-center, candidate gene studies, with limited replication across populations or transplant centers. To date, there are no large-scale, prospective, multi-center studies that have definitively demonstrated improved clinical outcomes from routine genetic screening in transplant recipients or donors. As such, these findings should be interpreted as hypothesis-generating rather than practice-changing. For example, genes like *CAV1*, *APOE*, and *VDR* have all been implicated in transplant outcomes [59,60,61], but results can vary significantly depending on the study population, definitions of rejection, and clinical protocols. Without standardized methods and larger validation studies, it is difficult to know how much significance to give to any single genetic marker.

Another challenge is that non-HLA effects tend to be subtle and multifactorial. Unlike HLA mismatches, which are often clear-cut, non-HLA variants usually have modest effects that depend on other factors like the donor age, ischemia time, or immunosuppressive regimen [63]. In SPK recipients, the picture is further complicated by the potential for both alloimmune and autoimmune reactions. Unlike recipients of other organs, patients with T1DM have an underlying autoimmune disease that can re-emerge after transplantation. Several studies have shown that SPK recipients can develop or retain antibodies against islet antigens such as GAD65, IA-2, and ZnT8, which may signal the early recurrence of diabetes or contribute to pancreas graft dysfunction [58,67]. This makes it difficult to distinguish whether immune responses are being triggered by the donor or are part of the recipient’s own autoimmune memory.

Non-HLA antibodies are another area of growing concern. Antibodies targeting antigens like MICA, as well as tissue-specific self-antigens in the kidney and pancreas, have been linked to graft inflammation and rejection [58,61,66,67]. However, we still lack standardized tools for detecting these antibodies and clear guidelines on how to respond if they are present in a patient without traditional signs of rejection.

Finally, there are practical barriers. The genetic screening of donors for non-HLA markers is not routinely performed, and most transplant programs do not yet have the infrastructure to interpret or act on this information in real time. However, emerging data suggest potential applications for personalized decision-making in transplantation (Table 2). For example, donor *CAV1* genotypes could inform organ acceptance decisions, particularly when multiple donor offers are available [59]. In addition, *CYP3A5* polymorphisms in recipients have consistently predicted tacrolimus metabolism; genotype-guided dosing could improve early drug-level targeting and reduce the risks of under- or overexposure [84,85]. Variants in immune regulation genes such as *TGF-β* or *IL-10* may eventually contribute to risk stratification algorithms to guide the intensity or duration of immunosuppression. While these applications remain investigational, this framework provides a rationale for integrating genetics into future clinical trials and prospective registries aimed at improving individualized care in SPK transplantation. Until we have obtained robust prediction models and validated protocols, non-HLA genetics will likely remain an important research area rather than a routine part of clinical care. In addition, this narrative review does not aim to advocate for the immediate clinical implementation of genetic testing, but rather to synthesize the existing literature and highlight recurring associations that need to be addressed through larger and more robust studies.

Another challenge might be the cost of genetic and polygenic risk profiling (PRS), since considerations of cost-effectiveness and health policy are increasingly relevant. Although the cost of genotyping has declined significantly in recent years, the integration of genetic testing into transplant workflows still encounters logistical and economic barriers, especially for multi-gene and PRS applications. At present, routine pre-transplant genetic screening is not the standard of care, and reimbursement pathways remain limited in many health systems. For future implementation, studies must be coupled with health economic analyses to determine feasibility, equity, and scalability across transplant centers.
genes-16-00884-t002_Table 2Table 2Clinical decision framework for genomic integration in SPK transplantation.Clinical StageGenetic MarkersClinical ApplicationDecision ImpactReferences1. Recipient Risk Stratification*HNF1A*, *HNF1B*, *INS*, *KCNJ11*, *GCK*Identify MODY/neonatal diabetes in atypical T1DM casesAvoid unnecessary SPK; manage with oral agents if C-peptide is preserved and autoantibodies are absent. Consider SPK if renal involvement is severe[38,39,40,41,42]
*TNF-α*Assess immune activation and rejection riskAdjust immunosuppression; consider thrombosis prophylaxis[32]
*F5*Evaluate thrombotic risk and rejection susceptibilityGuide perioperative anticoagulation and risk management[71]
*VDR* FokI FFAssess susceptibility to acute rejectionModify immunosuppressive therapy accordingly[61]
*APOE*Identify cardiometabolic riskIntensify cardiovascular monitoring; initiate lipid-lowering therapy[60]2. Donor Risk Stratification*CAV1* (rs3801995, rs9920)Predict fibrosis, ischemia–reperfusion injury, and immune reactivityGuide donor selection; personalize the immunosuppressive regimen[59]
*HO-1*Evaluate ischemia–reperfusion injury riskImplement protective measures during procurement and reperfusion[63]
*VDR* FokI FFAssess immune activation riskInfluence donor–recipient matching and immunosuppression[61]
*APOE*Identify dyslipidemia risk post-transplantInform metabolic monitoring and intervention[60]
MICADetect alloimmune response riskSupport immunological risk stratification[66]3. Monitoring and SurveillanceCD20, 34-gene AMR panelDetect antibody-mediated rejection noninvasivelyEnable early therapeutic intervention; reduce biopsies[73,74]
tCRM scoreQuantify rejection severity and treatment resistanceGuide treatment decisions and monitoring[75]
*GNMZ*, *PRF1*, HLA-DRMonitor T-cell activation and immune statusAllow the real-time assessment of rejection[78,79,80]
Donor-derived cfDNADetect allograft injury and rejection beyond 45 daysProvide high-sensitivity, noninvasive surveillance[81]
*MBL2*Assess innate immune function and allograft stressTailor follow-up intensity; consider immune modulation[76,77]
MICA/MICBIndicate early graft injury and immune activationAssist in timely clinical intervention[66]


## 6. Ethical Considerations in Pancreas–Kidney Transplantation

The integration of genomic data into clinical decision-making has the potential to significantly enhance outcomes in pancreas and pancreas–kidney transplantation. However, this emerging capability raises important ethical, legal, and social considerations that must be thoughtfully addressed to ensure responsible implementation [86].

From an ethical perspective, the use of genomic data in transplantation presents complex challenges. One of the most prominent is informed consent. Transplant recipients may be asked to consent to genetic testing for the purposes of immunological matching, pharmacogenomic profiling, or assessing predispositions to transplant-related complications [87]. Ensuring that patients understand the scope and potential downstream use of their genomic information is essential. Furthermore, the principle of beneficence must be balanced with non-maleficence, particularly when incidental findings or non-actionable variants are uncovered [88]. A key ethical concern involves the potential for genetic discrimination, where individuals with genetic markers associated with increased rejection risk or unfavorable drug responses could face inequality in their access to transplantation or post-operative care [89]. At the societal level, genomic applications in transplantation raise significant concerns about equity and access. Not all patients have equal access to advanced genomic testing and interpretation, which could exacerbate existing disparities in transplant outcomes based on socioeconomic status, geographic region, or institutional capacity [90]. The familial implications of genomic findings can be profound, especially if hereditary conditions are identified. This raises complex questions about the duty to inform family members, patient privacy, and the psychosocial burden of genetic knowledge [91]. This underscores the need for fairness and justice in genomic data use and organ allocation policies.

While genomics offers exciting possibilities to personalize and improve pancreas and pancreas–kidney transplantation outcomes, the practice of giving careful attention to the ethical, legal, and social dimensions is imperative. Policies and practices must prioritize equity, privacy, and transparency, ensuring that innovation does not come at the expense of fairness or trust in the transplant community.

## 7. Future Vision

Despite advances in surgical techniques and immunological management, graft pancreatitis remains a significant complication in pancreas transplantation, affecting 17–30% of recipients [92]. Understanding the molecular mechanisms underlying ischemia–reperfusion injury may enable the development of targeted therapies to improve surgical outcomes. Looking ahead, the integration of genetic insights into clinical practice holds promise for enhancing donor–recipient matching, personalizing immunosuppression, and improving graft survival. While HLA matching remains essential, emerging evidence points to the influence of non-HLA genetic factors, such as minor histocompatibility antigens, on graft rejection and longevity. Advances in genomic technologies may also enable genetic profiling to guide individualized immunosuppressive regimens and predict patient-specific risks, reducing rejection and improving long-term outcomes. Ultimately, leveraging genetics in pancreas transplantation could enhance risk stratification, support precision therapies, and extend graft function. The growing application of PRS in transplantation offers new opportunities to predict post-transplant complications such as skin cancer, allograft rejection, and long-term allograft failure [93,94,95]. Optimizing PRS models requires the integration of genetic data with clinical variables, such as immunosuppressive regimens and metabolic profiles, and validation across diverse populations. In the context of rejection and allograft dysfunction, donor–recipient genetic interactions may also inform risk stratification for post-transplant glucose dysregulation. Similarly, PRS for post-transplant skin cancer could benefit from large, multi-ethnic studies to improve calibration and combine genetic risk with environmental and clinical factors for personalized surveillance strategies. Beyond PRS, precision medicine is reshaping transplantation by tailoring therapies to the genetic, molecular, and environmental characteristics of both recipients and donors. While such approaches are more widely established in kidney transplantation, there is a growing need to extend them to pancreas transplantation. Advances in next-generation sequencing are enabling more nuanced donor–recipient matching beyond traditional HLA loci, potentially reducing rejection and improving long-term graft survival. Gene-editing technologies such as CRISPR/Cas9 have introduced the possibility of engineering donor organs to improve compatibility; recent successes with gene-edited pig kidneys in non-human primates signal early but promising steps toward clinical translation [96,97]. Initiatives like iGeneTRAiN are helping identify genetic variants associated with graft function and longevity, supporting the development of personalized immunosuppressive regimens and individualized post-transplant care [98]. The integration of PRS, gene editing, and personalized genomics marks a paradigm shift in transplantation, advancing donor–recipient matching and improving outcomes. Multi-omic strategies incorporating transcriptomics, proteomics, and epigenetics combined with artificial intelligence and machine learning, offer powerful tools for precision transplantation. However, these innovations must be accompanied by thoughtful ethical policy frameworks to ensure equitable access to genetic testing and avoid exacerbating disparities in organ allocation. Future research should focus on refining and validating predictive models, integrating multi-dimensional data into clinical workflows, and translating genomic insights into actionable strategies to improve long-term outcomes for transplant recipients and donors alike.

## 8. Conclusions

Genetic testing in pancreas–kidney transplantation is an emerging field with the potential to improve graft selection, predict long-term outcomes, and personalize immunosuppressive therapy. While HLA matching remains a cornerstone, growing evidence supports the role of non-HLA genetic variants, including gene deletions and polymorphisms, in influencing alloimmune risk and graft survival. These insights may be particularly valuable in optimizing the use of marginal donor organs and identifying recipients at elevated risk for rejection. However, clinical integration remains limited due to challenges such as the complexity of gene–environment interactions, the inconsistent replication of findings, and a lack of large, well-characterized cohorts. Additionally, the role of genetic testing in monogenic forms of diabetes leading to pancreas transplantation is still poorly defined. International collaborative efforts, including iGeneTRAiN, are helping to address these gaps by aggregating genome-wide data from thousands of donors and recipients [98]. These efforts support the identification of clinically relevant genetic variants and the development of individualized immunosuppressive protocols. In summary, genetic testing holds significant promise for advancing pancreas–kidney transplantation, with large-scale genomic initiatives playing a key role in translating discoveries into personalized care.

## Figures and Tables

**Figure 1 genes-16-00884-f001:**
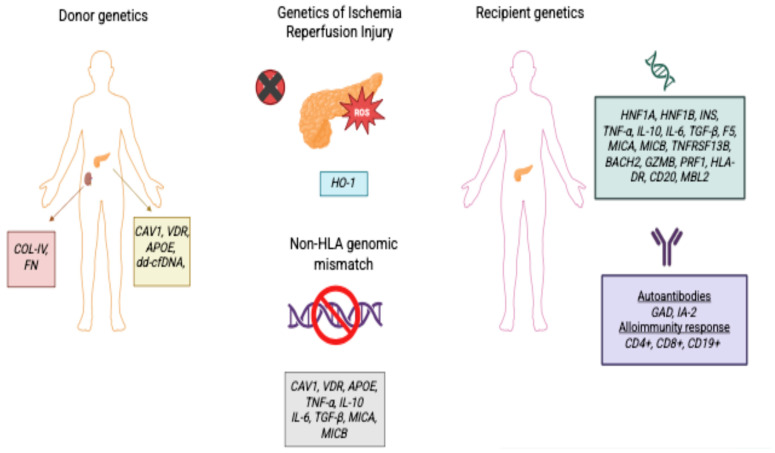
The illustration of non-HLA genetic factors in the recipient and donor contributing to the ischemia–reperfusion injury process, as well as the impact of genomic mismatches in pancreas and simultaneous pancreas–kidney transplantation. Created in BioRender. Caliskan, Y. (2025) https://BioRender.com/vy81q2b (accessed on 24 July 2025).

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
