# Peer review of "Genomics in Pancreas–Kidney Transplantation: From Risk Stratification to Personalized Medicine"

_genes, 2025, doi:10.3390/genes16080884_

Round 1
Reviewer 1 Report
Comments and Suggestions for Authors
Major issues
- The review reads like a long literature summary rather than a critical analysis. Multiple findings are mentioned without assessing the strength of evidence, limitations, or biases in cited studies. The reviewer recommends that the authors should incorporate evidence tables, summarize study types (e.g., cohort, GWAS), and appraise study quality.
- Many paragraphs repeat the same themes (e.g., TNF-α, IL-10 polymorphisms) across multiple sections (e.g., Section 2 and 3.1.2). And the sections are fragmented and jump between topics such as monogenic diabetes, polygenic risk, autoimmune recurrence, and biomarkers without clear thematic organization.
- No clear evidence shows routine genetic screening improves outcomes. Several studies cited are small or single-center. The authors should clearly distinguish between hypothesis-generating versus practice-changing findings.
Reviewer 2 Report
Comments and Suggestions for Authors
This is an interesting narrative review article that explores the role of genetics and genomics in pancreas and simultaneous pancreas-kidney (SPK) transplantation, emphasizing non-HLA genetic factors in graft rejection, survival, and personalized medicine strategies. It addresses both recipient and donor genetic influences, summarizes known biomarkers, and outlines future genomic tools like polygenic risk scores (PRS) and gene expression profiling. The paper covers monogenic and complex genetic disorders, with in-depth discussions of MODY, T1DM, T2DM, and how they influence transplantation decisions. The authors also evaluate both recipient and donor genetic contributions, adding a two-sided dimension rarely seen in transplant literature. The citations of the article include seminal studies and recent developments from major GWAS, the iGeneTRAiN consortium, and multiple peer-reviewed studies of genetic biomarkers. Table 1 is well referenced and summarizes non-HLA genetic variants and their clinical implications. The review highlights non-HLA mismatches (e.g., MICA/MICB, TNF-α, IL-10, APOE, VDR, HO-1) as important but underutilized predictors in clinical practice and the authors discuss autoimmunity recurrence in pancreas grafts—an often overlooked issue. It also reviews gene-editing tools (CRISPR), PRS, multi-omics, and machine learning as part of precision transplant medicine and emphasizes the need for diverse genomic cohorts and ethical equity in genomic access.
However, this article is a narrative review; while informative, it does not provide new statistical analysis, systematic review methodology, or meta-analysis. Although it lists clinical integration challenges, there's limited detail on cost, infrastructure, or bioethical concerns, which are critical for practical adoption of genomics in transplantation. Certain themes (e.g., TNF-α and IL-10 SNPs) recur multiple times in slightly different contexts, which could be streamlined for conciseness. Whereas table 1 is extensive and informative, it could benefit from better formatting (e.g., consistent use of abbreviations, clearer legends).
A few points to be addressed by the authors:
- Please add a clear clinical decision framework: How exactly might genetic data guide SPK candidacy, immunosuppression, or monitoring strategies?
- Please add more on cost-effectiveness and health policy: Integration of PRS and genetic testing into transplant workflows should include economic modeling.
- The paper would benefit from expanding on the ethical section; genetic data raises issues in consent, equity, and allocation that deserve more exploration.
Overall, this is a rich and timely review that effectively bridges genomic science and clinical transplant medicine. It succeeds in synthesizing current literature on non-HLA genetics in pancreas-kidney transplantation, introduces important emerging tools, and advocates for more personalized care. It is highly recommended for clinicians, transplant researchers, and genomic policy makers interested in the future of precision transplantation.
